# Effect of Climate Change on Introduced and Native Agricultural Invasive Insect Pests in Europe

**DOI:** 10.3390/insects12110985

**Published:** 2021-10-31

**Authors:** Sandra Skendžić, Monika Zovko, Ivana Pajač Živković, Vinko Lešić, Darija Lemić

**Affiliations:** 1Department of Agricultural Zoology, Faculty of Agriculture, University of Zagreb, Svetosimunska 25, 10000 Zagreb, Croatia; ipajac@agr.hr (I.P.Ž.); dlemic@agr.hr (D.L.); 2Department of Soil Amelioration, Faculty of Agriculture, University of Zagreb, Svetosimunska 25, 10000 Zagreb, Croatia; mzovko@agr.hr; 3Innovation Centre Nikola Tesla, Unska 3, 10000 Zagreb, Croatia; vinko.lesic@icent.hr

**Keywords:** climate change, agriculture, invasive species, invasive insects, biological invasion, insect pests

## Abstract

**Simple Summary:**

Invasive insects, along with climate change, are among the two most important environmental problems facing the world today. They pose a threat to many ecosystems worldwide, especially agriculture. As a result, there is a serious risk of economic losses to crops and a challenge to human food security. The aim of this review is to examine the relationship between climate change and the process of invasion of economically important insects in Europe. In recent decades, globalization has led to an increase in the worldwide movement of people and goods, resulting in an increase in the number of insects introduced into areas outside their original range. The harmful effects of invasive insects may be exacerbated by climate change as barriers to their successful establishment and dispersal decrease. To limit economic and environmental damage, it is important to understand the biotic and abiotic factors that influence the process of insect invasion in the context of climate change. We highlight the main biotic factors that influence the biological invasion process. Finally, we present the adaptive management strategies for invasion of non-native insect pests’ invasion that include prevention, eradication and assessment of biological invasion in the form of predictive modelling.

**Abstract:**

Climate change and invasive species are major environmental issues facing the world today. They represent the major threats for various types of ecosystems worldwide, mainly managed ecosystems such as agriculture. This study aims to examine the link between climate change and the biological invasion of insect pest species. Increased international trade systems and human mobility have led to increasing introduction rates of invasive insects while climate change could decrease barriers for their establishment and distribution. To mitigate environmental and economic damage it is important to understand the biotic and abiotic factors affecting the process of invasion (transport, introduction, establishment, and dispersal) in terms of climate change. We highlight the major biotic factors affecting the biological invasion process: diet breadth, phenological plasticity, and lifecycle strategies. Finally, we present alien insect pest invasion management that includes prevention, eradication, and assessment of the biological invasion in the form of modelling prediction tools.

## 1. Introduction

The problem of harmful invasive insects is becoming more serious and climate change is one of the main reasons that makes the problem of invasive insects even more complex for agricultural systems worldwide [1]. The phenomenon of climate change consists of the combined effects of increasingly variable and severe weather conditions combined with increased levels of atmospheric greenhouse gasses (GHGs), which are the main drivers of global warming [2]. The combined effects of the various drivers of climate change are exerting enormous pressures on agricultural systems and the food production industry. The greatest climatic pressure on crop production is the increasing frequency of extreme weather events, such as changing rainfall patterns leading to prolonged droughts or floods, and temperature fluctuations leading to heat waves and frosts [3]. These pressures have been divided into direct and indirect impacts: the direct impact of climate change on host plants and the indirect impact of climate change on insect pests and other harmful organisms that affect crop production [2]. Climate warming has a significant impact on the reproduction, survival, dispersal, and population dynamics as well as the relationships between pests, the environment, and their natural enemies [4]. In addition, a significant proportion of agricultural pests are alien or invasive insects [1]. Invasive species are organisms that are outside their geographic range that may (or already are) become harmful to human or animal health, the economy, or the natural environment [5].

For invasive insects, many researchers predict expanded geographic distributions, increased population densities, and voltinism under different climate change scenarios [6,7,8,9,10,11,12,13]. Invasive insects exhibit some common characteristics, such as a high reproductive capacity, high environmental tolerance and ecological adaptation, high insecticide tolerance, a broad host range, and competitive dominance [14]. Climate-related changes will lead to changes in the distribution of invasive species as their populations respond to fluctuations and changes in temperature, humidity, and biotic interactions [15]. They have a greater potential to be introduced from geographical regions with similar climatic conditions [16]. Moreover, if an invasive insect encounters unfavorable climatic conditions in a new geographic region, the probability of its establishment is low. However, climate warming has contributed to the increased likelihood of invasive pests becoming established in new areas outside their natural geographic range [13] by creating new ecological niches that provide opportunities for their establishment and spread [17]. It is important to note that climate change is not the only cause of biological invasions. The introduction of most alien pest species into a new geographic region may occur naturally or be mediated by international trade and human mobility [18]. Once an alien species arrives in a new habitat, the other stages of the invasion process (introduction, establishment, and dispersal) can be positively or negatively influenced by existing climate and ongoing climate change [8]. If climate change can alter the success of insect pest invasion with a noticeable impact on agricultural productivity, then characterizing these effects is of great importance for food security [19].

In this paper, we aim to present the effects of projected climate change on invasive insect pest species, the stages of their biological invasion, and their traits that determine the success of invasion. Possible measures to prevent and control such species are also described in the following text.

## 2. Climate Change and Its Impact on Insect Pests

Climate is a key element that affects various characteristics and distributions of managed and natural systems, including hydrology and water resources, cryology, marine and freshwater ecosystems, terrestrial ecosystems, forestry, and agriculture [20]. Climate change can be explained as the phenomenon that involves changes in environmental factors such as temperature, humidity, and precipitation over a long period of time [21]. Based on a number of global climate models and development scenarios, it is expected that the earth may experience global warming of 1.4 to 5.8 °C in the next hundred years [22]. The main cause of global warming is increased concentrations of greenhouse gases in the atmosphere. The most prevalent atmospheric gases are carbon dioxide (CO_2_), methane (CH_4_), and nitrous oxide (N_2_O) [23]. The rise of CO_2_ is one of the most important documented atmospheric changes of the last half century [24]. Its concentration has increased dramatically from 280 ppm in the pre-industrial era to 416 ppm and is expected to double by 2100 [22,25]. CO_2_ is considered a greenhouse gas because it strongly absorbs certain wavelengths of thermal infrared radiation emitted from the land surface. The more atmospheric gases absorb thermal infrared radiation from the Earth’s surface, the greater the fraction of radiation emitted from the atmosphere toward the Earth’s surface, resulting in an increase in air temperature [22,23].

From the Intergovernmental Panel on Climate Change (IPCC) report in 2013, the climate in southern Europe will become warmer and drier, while it will become warmer and wetter in northern Europe. With climate change, temperatures in Europe, especially at higher latitudes, are expected to increase more than the average warming worldwide [22]. Increased global average temperatures and varying precipitation patterns promote extreme natural events that affect landscapes and pose a major challenge to the natural environment, agriculture and food security [26]. Temperature is the major factor affecting plant and animal distribution and abundance patterns due to the physiological limits of individual species [27]. Ongoing climate warming allows the cultivation of more crop species but also favours insect pest survival and spread in these regions [28]. An increase in temperature can affect herbivorous insects directly through physiology, life cycle, or shift in geographic range and indirectly through the presence of host plants [6]. Many researchers have shown that an increase in temperature strongly influences insect pest dynamics. As temperatures rise, many insect pest species are expected to expand their geographic ranges from subtropical and tropical regions at lower altitudes to temperate regions at higher altitudes [27,29,30]. Climate warming could lead to a higher incidence of insect-transmitted plant diseases, as insect vectors also expand their geographic range and multiply rapidly under such conditions [31]. Milder winters could influence insect population dynamics via reduced mortality and increased population build-up [9]. Temperature increase accelerates insect development and reproduction rates, resulting in more generations per year and eventually more crop damage [6,32]. For example, Bergant et al. [33] attempted to quantify the potential impact of climate change on the development dynamics of Onion thrips (*Thrips tabaci* [Lindeman]) based on available climate change simulations using Global Circulation Models (GCMs), their projections to climatically heterogeneous sites in Slovenia, and a simple degree-day model. Their results show that the expected temperature increase will lead to a larger number of degree-days and a longer period with temperatures above the estimated lower threshold for *T. tabaci* development. Thus, the number of *T. tabaci* generations per season will increase due to the predicted increase in cumulative degree days and the lengthening of the period with favourable development conditions. More generations will lead to an increase in population and consequently more crop damage.

Finally, climate warming could favour the biological invasion process of non-native insect pests [34].

The documented increase in atmospheric CO_2_ concentration will change the biology of agricultural insect pests in two ways. The first change is related to climate stability and the second likely consequence is related to effects on host plant biology [19]. Numerous reviews indicate that current and projected increases in atmospheric CO_2_ concentrations will most likely stimulate the photosynthesis, reproduction, and growth of a wide range of plant species and act as plant fertilisers [35]. This indirectly affects insect pests by altering the quantity and quality of vegetation. With increased CO_2_ concentrations, the chemical composition of the leaves change, affecting the nutrient quality of the foliage and palatability to herbivorous insects [36]. Such plants accumulate sugars and starches in their leaves, which reduces palatability as the ratio of C (carbon) to N (nitrogen) changes [37]. Nitrogen is the key element in the insect body and therefore an increased CO_2_ concentration leads to increased plant consumption by some pest groups [38]. This can lead to increased plant damage as pests must consume more plant tissue to obtain the same amount of required nutrients. Increased consumption rates are a common response of leaf chewers such as caterpillars and leaf miners to a reduction in nitrogen content as predicted by CO_2_ fertilization, with compensatory feeding [39].

The frequency of precipitation has decreased while the intensity has increased. This type of fluctuating rainfall pattern has led to more frequent droughts and floods. Insects that overwinter in the soil are most affected by overlapping rainfall and prolonged stagnation of water. This threatens the survival of insects and negatively affects their diapause [21]. Drought affects pests indirectly through the host plant response. Drought-stressed plants are more susceptible to insect attack because of a decrease in the production of secondary metabolites that have a defence function [40].

As mentioned earlier, many herbivorous insects are invasive species and have a major impact on agricultural production. Changes, including altered climatic conditions and introduction mechanisms, facilitate the establishment and spread of insect pest species outside their original range [41]. In general, the fate of invasive insect pest species is highly dependent on the climate and its changes. The complexity of the combined interaction of invasive species and climate change is radically increasing, and confirmatory data on how climate change is exacerbating the already destructive impacts of invasive species are rapidly accumulating. The impacts of climate change, particularly temperature increases, may increase opportunities for biological invasions due to the adaptability of invasive species and the potential for a wider range of biogeographic conditions [42].

## 3. Invasive Insects

Invasive alien species (IAS) are species whose introduction or dispersal outside their natural past or present distribution range threatens biological diversity [43]. Biological invasions are considered to be important drivers of global change and are responsible for impacts on ecosystem functions, native species, and significant economic losses [44,45,46]. The increasing mobility of people and the international trade system has brought an increased possibility of the movement of different species around the globe, either intentionally in the form of commodities such as pets, livestock, nursery stock, or agricultural and forestry products [47].

It is likely that global climate warming will create suitable environmental conditions for the translocation of insects from tropical areas to temperate regions [48,49,50]. Establishing a permanent population in a new environment is easier for herbivores, who can adapt to the given conditions if a suitable host plant is available [6], and for polyphagous herbivores that are able to exploit new host plants [9]. Between 30 and 45% of all herbivorous insects in agriculture and forestry worldwide are considered alien or invasive insects [51]. Vilà et al. [45], using the DAISIE database, estimated that 24.2% of alien invertebrates in Europe have economic importance. Findings of quarantine pests are entered into the European and Mediterranean Plant Protection Organization (EPPO) central communication database [52]. The EPPO A1 list contains pests that are absent in the EPPO region, and the EPPO A2 list contains pests that are present in a limited geographic area of the EPPO region. There is also a third EPPO list—the Alert List, whose main purpose is to draw the attention of EPPO member countries to specific pests that may pose a risk to them and to provide early warning [53].

The European Commission has established a list of priority pests for European Union. Priority pests are Union quarantine pests that meet all of the following conditions: (I) their occurrence in the Union territory is unknown or they occur either only in a limited part of that territory or only rarely, irregularly and sporadically; (II) their potential economic, environmental, or social impact is the most serious in relation to the Union territory. The methodology for determining which pests are included in the priority list comprises composite indicators and a multi-criteria-based analysis. It takes into account, for the Union territory, the likelihood of the spread, establishment, and impact of the pests assessed [54]. Table 1 lists priority the arthropod pests that pose a threat to agricultural crop production in the European Union.

### Case Studies

Over 1000 insect species have already invaded Europe, including some of the most damaging invasive insects such as the Tobacco Whitefly (*Bemisia tabaci* [Gennadius]), the Western Corn Rootworm (*Diabrotica virgifera virgifera* [LeConte]) and the Colorado Potato Beetle (*Leptinotarsa decemlineata* [Say]) [52] which are included in the EPPO A2 list. Among the invasive insects listed on the EPPO A2 list, the South American Tomato Pinworm (*Tuta absoluta* [Meyrick]) is currently drawing worldwide attention as one of the world’s most damaging pests of solanaceous crops with a high ability to infect new geographical areas threatening tomato production worldwide [80,81]. A recent study showed that *T. absoluta* infested 60% of tomato crops worldwide during the period 2007–2017, with an increase in range of 800 km per year [82]. Santana et al. [81] presented a model suggesting that predicted climate changes will negatively affect *T. absoluta* in regions around the equator and positively in regions near the poles. More specifically, southern European countries will become unsuitable for *T. absoluta*, and on the other hand, most northern countries in Europe will become more suitable due to global warming.

One of the pests previously included in the EPPOs’ Alert list is an extremely damaging polyphagous insect, the Brown Marmorated Stink Bug (*Halyomorpha halys* [Stål]) [83]. Since the early 2000s, *H. halys* has spread throughout most European countries and has become a significant threat to agricultural production [84]. However, Kistner [7] showed that future climate change scenarios suggest that an increase in summer temperatures will reduce the growth potential of *H. halys* compared to current climate conditions, which in turn may reduce the damage to summer crops. He also showed that climate change could increase the number of generations produced annually, making the invasive insect multivoltine in the northern latitudes of Europe, where it is currently considered univoltine. The Fall Armyworm (*Spodoptera frugiperda* [Smith]) is a pest included in a EPPO A1 list. It is a highly polyphagous and destructive pest causing massive damage to crops, especially corn. While its dispersal is attributed to the global transport fruits and vegetables, its invasiveness is attributed to its high adaptation ability to different environmental conditions. With expected climate change scenarios, it is probable that this species will rapidly invade large areas of the world [71]. For a long time, the pest was present only in its native range in tropical–subtropical areas of the Americas [85,86]. According to EPPO [67] the first occurence of *S. frugiperda* outside its native range was recorded in Africa in 2016 and the first occurrence on the Asian continent was recorded in Yemen and India in late July 2018. In June 2019, the pest was recorded in Egypt and in 2020 in Jordan, which is the closest to Europe so far [67]. Overlapping generations, a large migratory capacity (adults were reported to fly 100 km per night) and a large number of host plants make *S. frugiperda* one of the most important pests threatening its introduction in Europe [69]. The European Food Safety Authority (EFSA) has carried out a risk assessment of the occurrence and domestication of *S. frugiperda* in the European Union. The study concluded that, according to current indicators, there is a possibility of its establishment in the Mediterranean region (Spain, Italy, and Greece) due to the increase in temperatures [87].

## 4. Phases of Biological Invasion by Alien Insect Pests

Invasions of non-native/alien species begin with the human-assisted movement of individuals or propagules of the species across biogeographic barriers [88]. The increasing global movement of people and goods is driving the increasing rate of biological invasions by alien insects [89]. In addition, trade flows, along with human travel, help alien insects to cross natural barriers, linking globalization and climate change to successful biological invasions [13,90]. The biological invasion process of alien insects comprises a chain of events including transport, introduction, establishment, and dispersal [91].

### 4.1. Transport and Introduction

Insects in various stages of life could be moved by natural means in the passive or active form of transport. Passive transport could be by wind, water currents, or animals, and active transport could be by the insects’ own movements [17]. Extreme weather events or changing circulation patterns may facilitate the introduction of some invasive insects into new geographic areas. Extreme climatic events such as hurricanes or storms could potentially transport insects long distances from their native region [41,92], and it is already known that climate change will increase the frequency and intensity of hurricanes [22]. Such insects could find favourable environmental conditions for its establishment in the new geographic region [17]. For example, the longest climate associated migration of an insect took place in 1988 when dessert locusts from Africa were located on Caribbean islands and the east coast of South America. A sub-tropical low pressure and the following hurricane carried locusts 4500 km [93,94].

Far more frequently, the transport of invasive insects is mediated by human activities via the international trade system and human travel [91]. Trade systems between Europe and North America have considerably unified fauna between these two continents [95] and therefore have increased homogeneity and thus disrupted the biodiversity of each. Thus, insects may cross a geographic barrier that previously defined the limitations of their historic range [91]. Despite the human-induced arrival of alien species, their establishment, dispersal, and biological success are altered by climate and its changes in a couple of ways [19].

Climate change may alter the patterns of human transport, changing the pressure of introduction of potentially invasive species. Invasion pressure could increase through new or increased transport between a native and a destination region, or through improved survival of propagules during the transport phase. In the first case, climate change may link geographic regions that were previously separate; in the second case, climate change may affect biological processes associated with transport [41]. Some insect species are more prone to be introduced and dispersed into new geographic regions than others, and some pathways favour the introduction of some alien insect species [96]. The number of arriving insect individuals is called propagule pressure [95], also known as ‘‘introduction effort’’ [97]. Propagule pressure describes a function of the frequency and number of individuals introduced into a new habitat [1]. During the phase of introduction, propagules must reach a stage where they can sustain a local population [98]. Initially, an insect species must cross large geographic barriers to reach its new territory. Their ability to pass through the transport phase depends on the speed with which individuals are moved from one area to another, as well as their viability once they arrive. It is likely that insect species with the potential to invade will arrive if climate change alters the species’ potential to be successfully transported. Then, individuals must survive and withstand the new environmental conditions in their new territory. After that, individuals must gain food resources, persist in interactions with natural enemies, and even establish symbiotic relationships in the new area [41]. It should be noted that climate change influences phenological changes in host plants as well as natural enemies, and thus may also affect the success of insect invasion [99,100]. Recognizing potential herbivorous invaders and their pathways of introduction helps prevent or reduce the number of introductions because not all introduced insects are invasive, nor do they all succeed in establishing themselves under new environmental conditions [95]. Propagule pressure is related to the magnitude of the plant trade, the likelihood of alien insects being transported on these plants, and the likelihood of them passing border controls undetected in plant commodities [101]. One of the most recent examples of such a pathway is the case of the invasion of the highly polyphagous and harmful invasive insect, the Spotted Wing Drosophila (*Drosophila suzukii* [Matsamura]) in European countries. Trade in fresh fruit from its native range, South East Asia, is considered the main pathway of invasion, with the first individuals appearing unnoticed in large numbers at the egg or larval stage [102,103]. Crop losses due to *D. suzukii* infestations have been measured as up to 20% of crop yield from preferred hosts such as blueberries, strawberries, cherries, blackberries, and raspberries [104]. The first and most important task for introduced insects is to find a suitable host plant on which to feed [105], so generalist species such as *D. suzukii* are most likely to become invaders with such diet breadth [95]. After its initial discovery in Europe in 2008 (parallel reports from Italy and Spain) *D. suzukii* rapidly colonised most countries in Europe [96,106]. The risk of invasion may increase if shorter, warmer winters open northern regions to earlier and longer growing seasons and overwintering opportunities for *D. suzukii* [6,104].

The intensity and patterns of introduction are changing more rapidly today than ever in human history [107]. Stowaway transportation on passenger planes is an important and growing source of IAS introduction, as is transport by ship, making seaports the main epicentres of invasion [108]. As an example, *H. halys* is a pest species that is capable of being transported over very long distances by sea freight, ground transport vehicles, and air transport [83]. Furthermore, both nymphs and adults have a strong dispersal ability [109]. Climate change is also likely to further complicate the management of such a species, as the geographic distribution of *H. halys* is primarily determined by the climate and its disruptions [7]. Predicted rising temperatures could promote the development and survival of *H. halys*, especially in northern latitudes where a disproportionate increase in minimum winter temperatures is predicted [7,110,111]. Thus, under climate change scenarios, Europe will become more vulnerable to the establishment and spread of *H. halys* [7].

Although the initial introduction stage is a critical part of invasion success, it remains the most poorly studied stage of invasion because of the difficulty in detecting early propagules which hinders the effective management of early-stage invasions [90,112]. Therefore, there is a need for a more coordinated defence against invasions of such highly damaging pests in Europe [90]. Even if climate change causes the extinction of some invasive species and the advance of others, and the abundance of invasive species remains the same, it would be helpful to determine which species are predisposed to change [41].

Propagule pressure is dependent on the spread abilities of herbivorous insects, the distance they have to proceed and the environmental conditions of the habitat that they are invading [1]. The origin, identity, and volume of introduced insect species arriving into a new area could change significantly with climate change. Large scale shifts in the geographical patterns of agricultural production can be expected and therefore, the origin of agricultural commodities and its transport pathways may modify. This could allow a new way in which potential alien species can be introduced and make the most of each mode of transport pathway [1]. All things considered, it is evident that climate change will have a positive effect on propagule pressure of invasive insects and that we can expect many new introductions.

### 4.2. Establishment

Establishment is the process of forming a permanent population in a new geographic location different from the species’ place of origin. Numerous insect species are introduced in a new environment so often and in high numbers that they appear to be established, but they are still incapable of forming a stable population [95]. It is estimated that about 10% of accidentally introduced alien herbivorous insects successfully establish themselves, and 20–40% of purposely introduced biocontrol agents (predators and parasitoids) establish persistent populations [113].

Climate change may facilitate the establishment of introduced invasive species through two mechanisms: (I) species that are currently unable to survive in a new environment due to climatic limitations may increasingly be able to survive and become established there; (II) invasive species that can tolerate new climatic conditions may have a higher potential to overcome biotic constraints and establish permanent populations under climate change. Climate change can be expected to displace native species from conditions to which they are adapted, potentially making them less competitive [114]. If climate change increases the competitiveness and spread rate of non-native species, they may become established. The “lag phase” in invasions, in which species that establish small, apparently non-invasive populations shortly after introduction later become successful colonisers, is well known [41,115].

Once an alien insect is introduced, the chance for its establishment and persistence in the novel area depends on a broad range of biological (biotic) and environmental (abiotic) factors [17]. Biotic factors represent a set of different insect traits that may be useful in establishing a population in a new habitat. Insects that possess these traits can be favoured by climate change and may pose a significant problem for agriculture in the future. Ward and Masters [1] presented the main insect traits that are crucial for invasion success. These are: diet breadth, phenological plasticity, and lifecycle strategy adaptation.

Diet breadth is one of the most important insect traits linked to invasion success [116]. The first and most important task for introduced insects is to find a suitable host plant to feed on [105]. If a native or primary host in new habitat is not available, then it must seek a taxonomically related one and adapt to it. The suitable host must be sufficiently abundant for introduced insects to find them and must enable larval development and often maturation feeding of adults [117]. Rigorous host plant preferences could limit the chance of finding a suitable host in a new area, making generalist or polyphagous insects more presumable invaders [95] and climate change may make them even more successful [1]. For generalist species such as *D. suzukii*, which show extraordinary plasticity in their choice of food with over 80 host plant species, diet breath is one of the most important traits responsible for their successful colonisation [118,119]. This theory can also be supported by the fact that a large number of plant species are expanding their distribution worldwide as a result of climate change, making them available to herbivorous insects in search of their food. Although generalist species are considered winners in the race for habitats altered by global warming, Betzholtz et al. [120] have recently identified a number of traits that allow specialist butterfly and moth species to cope with climate change. Swedish butterflies and moths whose diet of young larvae is rich in nitrogenous plants have expanded their northern range limits faster than specialists that prefer other diets. These nitrogen-favoring insects have benefited from the combination of microclimate warming and heavy eutrophication that has occurred in 50% of habitats in Europe.

As poikilotherms, insect species are highly temperature-dependent, which is why they respond to temperature fluctuations with changes in their phenology. A large majority of phytophagous insects depend on close synchrony with their host plants and phenological synchronisation of insects and their host plants could be crucial for the establishment of introduced insects [105]. Global warming-induced advances in phenology are one of the best-documented biological responses to anthropogenic driven climate change [121]. Plants, fungi, and insects in a wide variety of temperate ecosystems have shifted occurrences of their biological activities to match an earlier beginning [49] and a later ending of the growing season [122,123]. Climate change leads to mismatches to the trophic relations in interacting species and results in climatically driven decoupling [8]. Decoupling occurs when the synchrony between species becomes disrupted in time and/or space [124]. Sensitivity to phenological change is likely to have a negative impact on spring feeding insects [125]. Species whose biological activity, such as egg hatching, is timed to the bud burst of host trees may be susceptible to any alteration to synchrony. These kinds of species usually have a narrow window of opportunity to maximise growth because they are constrained by starvation if they emerge too early and are also constrained by the declining nutritional value of maturing leaves [8,126]. The effects of phenological asynchrony are best known for the Lepidoptera species. One example is the Gypsy moth (*Lymantria dispar* [L.]) developing on black and red oaks (*Quercus velutina* [Lam.] and *Q. rubra* [L.]). Egg hatching before budburst will lead larvae to starvation, particularly in the absence of any alternative host plant. Conversely, if egg hatching happens too long after budburst, the quality of foliage could decline resulting in reduced fecundity [127]. The same mechanisms that lead to phenologic asynchrony and decoupling could allow some insects to utilise hosts plants that were formerly outside of their phenological range as climate change differentially modifies the seasonal timing of host plants and phytophagous insects [8,128]. Insects develop a variety of strategies to keep themselves alive under thermally stressful environmental conditions. Some of them use behavioural avoidance through migration and physiological adaptations such as diapause [28]. Knowing the terms of overwintering biology and the cold tolerances of potential invasive insect pest species would indicate the chances for their survival in the new area [129]. A majority of insects that come from temperate areas have some form of winter diapause [130]. Diapause is usually obligatory in univoltine species and facultative in multivoltine species in which diapause may be initiated as a response to biotic or abiotic triggers. The exact nature of these triggers could determine the response of insect species to climate change [1]. Temperature rises will have direct consequences for insects in the terms of their life cycle duration (developmental rate) and changes in voltinism. Many invasive insects that have broadened their geographic range exhibit latitudinal gradients in voltinism and therefore, altering temperatures should define the boundaries between voltinism states in predictable ways. Voltinism can be relatively plastic, and in some species, it is impacted by the temperature, photoperiod, and host plants, while in others it is fixed [131]. For those insects with flexible voltinism, temperature rises could be beneficial, allowing for faster growth and additional generations per year [6]. Increased voltinism could promote faster population growth due to the greater number of offspring produced annually and therefore increasing the chance for insect pest outbreaks or elevating non-pests or minor pest species to a higher economically important status [132]. Moreover, insects with asexual reproduction often have a superior ability to colonise and disperse rapidly, due at least in part to the ability of single individuals to establish new populations, the lack of need to find mates, and the approximately twofold advantage in population growth rates [133,134].

For the Green Peach Aphid (*Myzus persicae* [Sulzer]), climate warming is directly associated with increased population densities, higher development rates, and outbreak frequencies. Combined with the lack of corresponding effects on the natural enemies of *M. persicae* and the resulting reduced predation pressure, temperature increases lead to increased economic losses overall [28].

Another example of a species with voltinism changes and higher developmental rates across latitudes is *L. decemlineata* [135,136]. Moreover, the mean temperature rise associated with climate change has been observed to increase the range of that species in temperate regions because of the ecological release of thermal constraints and due to lengthening of the growth season. It has been shown that *L. decemlineata* can adaptively synchronise its life cycle with new environmental conditions and as long as the plant host is cultivated, climate change is likely to have beneficial effects on it [137,138]. Jönsson et al. [139] showed that a moderate increase in temperature will alter the voltinism of *L. decemlineata* in much of Europe. Some European regions that now have one generation per year may experience an increased frequency of two generations per year in future climate change scenarios. Furthermore, central and southern Europe will even experience a third generation of this economic pest species [140,141]. An additional generation may affect population dynamics, and a larger population at the northern limit of distribution will increase the risk of colonisation of currently uninfected areas [139]. By arriving in a new distant area, introduced insects face the new physical environment (climate and landscape) with abiotic constraints which could determine their invasion success. To become successful invaders, they have to adapt to it [104]. In general, a good match between physical conditions of the donor habitat and the new habitat of an alien insect will increase the chance for successful establishment. This match between areas is called biogeographical similarity [94]. An incapability to tolerate the physical and environmental conditions in a new area is one of the major reasons for the extinction of the founder population of alien insects [141].

### 4.3. Dispersal

The biological effects of climate change include modifications in the distribution of insect species and changes in the abundance of species within existing distributions that cause direct physiological effects on individual species, changes in abiotic factors, altered reproductive potential, and altered interactions among species [142,143]. Dispersal involves population growth and the dispersal of individuals of a species [144]. As densities increase, insect species will disperse and expand their geographic range to new areas with suitable environmental conditions [145].

Climate change may create more suitable conditions for the establishment and dispersal of invasive insects [47]. Invasive species are generally considered to have a wider range of tolerance or a wider bioclimatic envelope than native insects, allowing alien insects to find a wider range of suitable habitats [16]. A temperature shift can also have significant effects on a native species but little effect on an alien species, changing the competitive dynamics between them [146]. For pest species in general, a poleward shift in distribution limits associated with climate change is generally predicted [144]. In addition, a global temperature increase will potentially create suitable conditions for insects to disperse from tropical to temperate regions [48]. This could be an easy task for herbivorous insects, which can adapt to new environmental conditions if suitable host plants are available [6], and for polyphagous herbivores, which can use new host plants [9,147].

For herbivorous insects, host plant availability may be the most important determinant of the upper limit of their geographic range [148]. For example, climate change may affect the dispersal of *L. decemlineata* by altering the abundance and dispersal of its cultivated and non-cultivated host plants. The ranges of cultivated host plants can be affected by climate warming, as has been observed for the potato as the main host plant for *L. decemlinetata* in Europe [149]. The most important non-cultivated host plant of *L. decemlineata* is Buffalobur (*Solanum rostratum* [Dunal]), which is also a globally invasive weed species. It is native to Mexico and the United States, but currently thrives in Europe, Asia, and Australia [150]. Therefore, the spread of these two species, a weed and an herbivorous insect, may interact in invasions. Climate change could affect the spread of *L. decemlineata* by increasing the distribution and abundance of *S. rostrum*. Therefore, it is necessary to study the current and future global distribution of both species under the influence of climate change [151].

At higher elevations and latitudes, a shift in the species range may occur as temperatures rise and introduced species migrate from previously warmer climates [49]. Assuming that tropical systems will experience an even greater increase in temperature, they do not face the same risk because there is no pool of species that originates from even warmer climates. However, changes in rainfall patterns and other climatic variables could still stress tropical ecosystems and increase their vulnerability to invasive pest species [41].

A rise in temperature could increase the flight time of insects, allowing them to disperse over greater distances [13]. For example, the migration patterns of the Migratory Silver Y Moth (*Autographa gamma* [L.]) to the UK are significantly influenced by changes in temperatures and precipitation patterns in their overwintering areas (North Africa) [13,152]. Another example of successful dispersal associated with global warming is the walnut pest native to North America—the Walnut Husk Fly (*Rhagoletis completa* [Cresson]). It has crossed the Swiss Alps and established populations throughout Switzerland and other neighbouring European countries [50].

Pressure from natural enemies on insects is higher at lower elevations [153,154]. Consequently, changes in the demography and distribution of hosts and natural enemies will influence insect biotic interactions and their responses to climate change across elevational gradients [155,156]. When an invasive insect enters a new habitat without its natural enemies, it typically benefits from some type of ecological release that allows invasive insects to reach higher population densities than could occur in their native range [47].

## 5. Invasion Management in Terms of Climate Change

Management plans and climate change adaptation measures require the inclusion of invasive species management as an important tool for reducing pressure on key ecological services and improving ecosystem resilience. Determining potential changes in insect distribution due to climate change is an important guide for invasive species management and control [157]. The biological invasion of insect pest species triggered by climate change is influenced by existing levels of ecosystem resilience. The main factor of resilience is the ability of an ecosystem to tolerate a certain level of change while maintaining functionality [158]. Resilient natural or agricultural ecosystems are expected to remain functional even when affected by high levels of disturbance [50,159]. The main specific actions that can prevent the negative impacts of climate change and invasive insect pests are: preventing the introduction and establishment of new alien species to minimise the possibility of their subsequent impacts;eradicating or controlling existing invasive pest species (including harmful native species) that have the potential to fundamentally alter ecosystem composition and services, thereby increasing ecosystem resilience;assessing biological invasion potential in the context of planning and constructing adaptation measures, particularly those that serve to meet important human needs, such as agricultural practices [42] (Figure 1).

### 5.1. Prevention

Given the greater frequency of pest invasions, their negative impacts, and the significant resources required to rapidly control spreading alien species once they become established, the most cost-effective management strategy is prevention. Prevention involves controlling the introduction or establishment phase of a species. Preventing the introduction of alien insects begins with identifying and controlling key transport vectors and pathways [160]. Control of vectors and transport pathways involves rigorous inspection, enforcement, evaluation, and (if necessary) refinement [91]. Information on transport routes is fundamental for risk assessments, monitoring, surveillance and management of alien species [146]. In addition, prevention strategies that consider dispersal pathways, together with protocols that focus on the arrival of individual alien species in a given region, are helpful in reducing their negative impacts [89]. The international response to invasive species has been through agreements such as the International Plant Protection Convention (IPPC) of the Food and Agricultural Organization of the United Nations, the World Trade Organization Agreement on the Application of Sanitary and Phytosanitary Measures (SPS) and the Convention for Biological Diversity (CBD) [52].

Due to the uncertainty of the establishment of invasive species, whose success correlates with propagule pressure, prevention through border security measures and quarantine is the most effective management approach. This is especially true for invasive species that are considered high risk to a particular country or region [161]. The defence against invasion by regulated insects in the EPPO region is controlled by phytosanitary inspections at European borders [52]. Increase in travel and trade leads to a higher risk of insect invasion, but to combat this threat, detection technologies need to be further improved, as do control and surveillance techniques [161,162]. In their work, Bacon et al. [52] incorporated information on border inspections into a metric they called “trade volume to be inspected”, which was calculated from information on commodity, origin, quarantine insects, destination, and trade volume, such that transport routes with a higher risk of introducing quarantine insects had a high inspection value and a higher number of interceptions. This was used to identify inspection gaps in border control systems between European countries. They found that the countries with the largest inspection gaps or the poorest border controls also had the highest numbers of established alien insects.

### 5.2. Eradication

The CBD directs member nations to “prevent the introduction of, control or eradicate those alien species which threaten ecosystems” [43]. Eradication represents the removal of an alien population which should lead to the recovery of the formerly threatened area. Multiple conditions must be met for a successful eradication program [163,164]: (1) the target alien species must be detected early; (2) the biology of the alien species must be vulnerable to control measures; (3) eradication resources must be sufficient to complete the program; (4) managers must have the public support and authority to take all required steps; and (5) re-invasion must be prevented. To prevent the re-invasion of alien species, governments may impose quarantine measures to restrict the movement of certain goods or require that imported materials undergo treatments such as fumigation. Cargo and passengers arriving at ports of entry or border crossings may be screened to prevent the accidental introduction of pest species. Despite the best efforts of governments, these measures can never prevent the introduction of all invasive insect species. The continued increase in global trade and passenger flows may overwhelm containment measures [164].

Despite the desired goal of preventing or eradicating invasive insects, this is an enormously difficult task that will not succeed in the majority of cases. In most cases, managing invasive insects and controlling the rate of spread are the only options [134,165]. Eradicating and controlling invasive species is also a very costly process. For example, in New Zealand alone, USD 800 million is spent annually on invasive phytophagous insect species, which includes biosecurity measures such as monitoring, control and research [46,166]. There are no resources available to screen all incoming trade for potential invaders or to launch an eradication campaign for every invasive species as soon as it is detected. Therefore, eradication or control efforts must be prioritised to minimise the damage caused by a species that has been determined to have the potential to spread over a large area or have a serious negative impact on ecosystems [46].

### 5.3. Assessment of the Biological Invasion

An important first step in assessing the negative impacts of invasive pests is to predict temporal and spatial changes in the population of invaders. Common methods for predicting the geographic distribution of species are simulation models. Modelling tools have helped improve early detection and understanding of the dispersal dynamics of certain invasive species and have facilitated management strategies to slow their overall spread [162]. Various models are used to categorise potentially invasive species. These models can be divided into three broad categories: (I) comparison of climate factors and geographic ranges of invasive and native species to identify their potential as invaders e.g., CLIMEX [19,164]; (II) evaluation of shared characteristics of taxa within organisms known to have invasive potential; and (III) a “risk assessment method” that evaluates intrinsic and extrinsic factors in relation to biological invasion success [146].

Understanding the factors that influence the distribution of species is essential for researchers to develop effective adaptation strategies to address the impacts of climate change [167]. Studying the distribution of the niches of pest species reveals future opportunities and risks under climate change and helps in the development of effective crop management strategies [168]. Species Distribution Models (SDMs) are scientifically proven tools for predicting and assessing the impacts of climate change on flora and fauna [169,170]. These modelling tools can be used to determine the relationships between species and their environment and predict their distribution based on occurrence data [171]. Early detection and prevention increasingly rely on SDMs that predict suitable habitat on which to focus monitoring and eradication efforts [172]. Among the many tools available for creating SDMs, the Maximum entropy method (MaxEnt) is the most commonly used because it has been shown to be effective for most taxa [173]. MaxEnt is a general-purpose machine learning method used to generate species distribution maps using presence-only data [174].

MaxEnt contrasts environments of occurrence points with those of background locations to determine which combinations of variables best predict occurrence [175]. For example, Qin et al. [63] estimated the impact of climate change on the global distribution of the Oriental Fruit Fly (*Bactrocera dorsalis* [Hendel]) using the MaxEnt model in their study. Their result showed that under future climate conditions, suitable areas for *B. dorsalis* in the Northern Hemisphere would expand northward, while suitable areas in the Southern Hemisphere would expand southward. Both suitable areas and habitat suitability would increase, especially by 2070 under the RCP85 climate change scenario, when almost 20% of Europe will be climatically suitable for the species, especially parts of the Mediterranean coastal regions.

Furthermore, the use of modeling tools has great practical importance for local risk assessments of such economically relevant pest species and for the management of highly dispersive insects in general. Modelling tools can help land managers and policy makers make informed decisions about the uncertainties and risks associated with the control of alien species. Looking ahead, a greater investment of time and energy in collaborative analytical decision making processes can help managers identify the most cost-effective prevention methods and eradication measures [176].

## 6. Conclusions

The introduction of invasive insect pests poses a major threat to various ecosystems, biodiversity, and economies worldwide. The rate of introduction of such species will continue to increase with the international trade system and human mobility. Economically, the current impact of invasive insect pest species in agriculture is measurable in billions of dollars annually due to loss of productivity. The response of invasive insects to human-induced climate change is a topic that still requires much research. However, it is now known that climate and climate change affect invasive insects in many ways. First of all, it is an important abiotic factor that affects all stages of biological invasion. Climate change could induce different insect traits that could increase the potential for invasion success. It is believed that climate change and global warming could be even more beneficial to invaders in the future. Therefore, it is advisable to develop climate change adaptation plans and measures that will prevent the introduction and further spread of invasive insect pests. The most important specific measures that can prevent the negative impacts of climate change and invasive insect pests are the prevention, eradication, and assessment of biological invasion in the form of predictive modelling tools.

## Figures and Tables

**Figure 1 insects-12-00985-f001:**
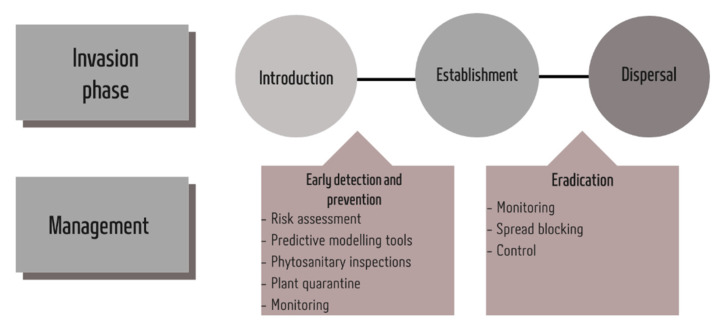
Schematic representation of strategies for the management of pest invasions.

**Table 1 insects-12-00985-t001:** Priority list of agricultural insect pest species for the European Union.

Specie	EPPO Categorisation	Distribution in EPPO Region	Host Range	Factors Contributing to Its Invasive Potential
*Anastrepha ludens*	A1 list	Absent [55]	polyphagous (*Citrus* spp., *Mangifera indica*, *Prunus persica*, etc.) [55]	- high availability of host plants [55] - transport of fruit containing live eggs or larvae [55] - flight dispersal [55]
*Anthonomus eugenii*	A1 list	Absent [56]	Solanaceae (*Capsicum* spp., etc.) [56]	- transported as immature stages in fresh fruits [57] - adults can survive prolonged cool conditions (2–5 °C) for over 3 weeks [57]
*Bactericera cockerelli*	A1 list	Absent [58]	Solanaceae [58]	- trade of plants of Solanaceae family [58] - flight dispersal of adults [59]
*Conotrachelus nenuphar*	A1 list	Absent [60]	*Prunus* spp. [60]	- human-assisted pathways: 1. as pupae in soil alone or in association with plants for planting 2. as overwintering adults in litter in association with plants 3. as adults in packaging material used to transport plants or fruits [60]
*Bactrocera dorsalis*	A1 list	Absent [61]	highly polyphagous on fruit species [62]	- trade of infested fruit [61] - climate warming [63]
*Rhagoletis pomonella*	A1 list	Absent [64]	Rosaceae (*Malus* spp., *Prunus* spp., etc.) [65]	- trade of infested fruit [64] - high availability of major host plants [64]
*Aromia bungii*	A1 list	Germany, Italy, Russia (Far East) [66]	*Prunus* spp. [66]	- transport in wood packaging [66] - flight dispersal [66]
*Spodoptera frugiperda*	A1 list	Israel, Jordan, Spain (Canarian Islands) [67]	highly polyphagous (Poaceae, Asteraceae, Fabaceae, etc.) [68]	- high reproductive potential [69] - strong flight capacity [69] - accidentally transported as contaminants of traded commodities [70] - highly adaptable to different environments [67] - climate warming [71]
*Thaumatotibia leucotreta*	A1 list	Israel [72]	polyphagous (*Capsicum* spp., *Citrus* spp., *Prunus* spp., *Gossypium hirsutum*, etc.) [73]	- trade of agricultural products from Africa [72]
*Bactrocera zonata*	A2 list	Israel [74]	polyphagous (*Citrus* spp., *Mangifera indica*, *Prunus persica*, etc.) [75]	- transport of infested fruits [74]
*Anoplophora chinensis*	A2 list	Croatia, Italy (both under eradication) [76]	polyphagous on woody hosts [76]	- problematic detection due to infestation, which may remain undetected for many years [76] - high availability of host plants [77]
*Popillia japonica*	A2 list	Italy, Portugal (Azores), Russia (Far East), Switzerland [78]	highly polyphagous [79]	- international trade [78] - flight dispersal [79]

## Data Availability

No new data were created or analyzed in this study. Data sharing is not applicable to this article.

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
