# Peer review of "Effect of Climate Change on Introduced and Native Agricultural Invasive Insect Pests in Europe"

_insects, 2021, doi:10.3390/insects12110985_

Round 1
Reviewer 1 Report
It is very difficult to follow the the changes authors made according to the comment of the Reviewer 1, Response 1 by the authors the L47- L50 is different than is written in the Response 1. This applies to all other responses from the authors. Only there is a text addition in Section 3. Invasion insects written in red. The main changes done is on the numbering of the citations inserting new reference numbers using red colour. .
Reviewer 2 Report
After carefully reading the corrected article, I find that the authors tookinto account all the comments of the reviewers. I find that the corrected
version of the article is suitable for publication.
Reviewer 3 Report
I am happy that the authors have addressed the reviewer comments.
This manuscript is a resubmission of an earlier submission. The following is a list of the peer review reports and author responses from that submission.
Round 1
Reviewer 1 Report
Climate change and invasive species are two of the most important environmental issues today. Although I am not the expert in the field of climate change and invasive insect, I wanted to get answers about following questions, (i) What impact would be the climate change on invasive insects? for each stage of invasion process? and why? (ii) What should we do in the future work to manage or eradiate the invasion species? Unfortunately, I didn't get a clear answer from this review paper.
In the introduction part, I think it would better to write ‘the problem of invasive insect is getting more serious’ first, and then write ‘climate change is one of the major reasons to make the invasive insect problem more serious or more complex’.
In the second paragraph of the second part, the first sentence said ‘increased temperature and varying precipitation promote extreme natural…’, but in this paragraph, just focused on the increased temperature, no effect of varying precipitation was described. In next paragraph, the increased CO2 will change the biology of insect pests in two ways. No description of how CO2 change the climate stability and climate stability influence the insect pests. Why ‘under such conditions, herbivorous insects tend to consume more plant material’? I cannot follow these sentences.
In other parts, this paper has too much texts on the introduction of invasive species, not on the impact of climate change on the invasive species, especially in the part of ‘invasion management in term of climate change’. I suggest the authors reduce some texts on the general description of invasive species, and focus and emphasize the impacts of climate change on the invasive species.
Author Response
Reviewer #1:
Dear reviewer,
I would like to thank you for your time and effort in reading our article. Also, for your comments and suggestions.
RESPONSE TO REVIEWER’S COMMENTS:
Point 1. What impact would be the climate change on invasive insects? for each stage of invasion process? and why?
Response 1: The problem of invasive insects and the stages of their invasion/colonisation are exacerbated by climate change. Climate change affects insects through a number of mechanisms. In the case of invasive insects, climate change favours strong biological traits that allow them to spread and invade intensively. These traits are described in our review, as well as their interaction with climate variability.
Additional lines added. Now L55 – L58
Point 2. What should we do in the future work to manage or eradiate the invasion species?
Response 2: The first task is to understand the factors affecting potential changes in distribution due to climate change. The ability to accurately predict how the distribution of invasive species and their impacts will change under projected climate scenarios is essential for developing effective prevention and control. It is well known that climate variables can influence the occurrence, distribution, reproductive potential, and survival of both native and alien species. Common methods for predicting the geographic distribution of species are simulation models. Modelling tools have helped improve early detection and understanding of the dispersal dynamics of certain invasive species and have facilitated management strategies to slow their overall spread.
Additional lines added. Now L640 – L652
Point 3. In the introduction part, I think it would better to write ‘the problem of invasive insect is getting more serious’ first, and then write ‘climate change is one of the major reasons to make the invasive insect problem more serious or more complex’.
Response 3: Point changed in the text. Now L27 – L29
Point 4. In the second paragraph of the second part, the first sentence said ‘increased temperature and varying precipitation promote extreme natural…’, but in this paragraph, just focused on the increased temperature, no effect of varying precipitation was described. In next paragraph, the increased CO2 will change the biology of insect pests in two ways. No description of how CO2 change the climate stability and climate stability influence the insect pests. Why ‘under such conditions, herbivorous insects tend to consume more plant material’? I cannot follow these sentences.
Response 4:
(a) Description of how increased atmospheric CO2 changes climate stability inserted. Now L98 – L105
(b) Explanation of how elevated CO2 affects herbivorous insects added. Now L150 – L160
(c) Explanation of how altered precipitation pattern effects on insect pests added. Now L160 – L167
Point 5. In other parts, this paper has too much texts on the introduction of invasive species, not on the impact of climate change on the invasive species, especially in the part of ‘invasion management in term of climate change’. I suggest the authors reduce some texts on the general description of invasive species, and focus and emphasize the impacts of climate change on the invasive species.
Response 5: Introduction has been revised and text non-related with climate change was removed.

Reviewer 2 Report
The manuscript ‘The impact of climate change on agricultural invasive insect pests in Europe’ by Skendžić et al provides a review into the effects of climate change on invasive pest insects, covering the different phases of invasion and management plans in the face of global climate change. This is a thorough review and all the information is supported using a variety of detailed case studies. It is well written and was an enjoyable read. I have a few minor points below:
The word ‘the’ is missing in a few places e.g.
L142: the new environment
L152: the European Commission
L182: the Alert list
L185: the European Union
L255: the new geographic range
L144: become an economically important pest (an missing)
L263: cross, not across
L317: species, not species (species is both the singular and plural form)
L441: insect pest outbreaks (pests does not need to be plural because outbreaks is)
Author Response
Reviewer #2:
Dear reviewer,
I would like to thank you for your time and effort in reading our article. Also, for your comments and suggestions.
RESPONSE TO REVIEWER’S COMMENTS:
Point 1. I have a few minor points below:
The word ‘the’ is missing in a few places e.g.
L142: the new environment
L152: the European Commission
L182: the Alert list
L185: the European Union
L255: the new geographic range
L144: become an economically important pest (an missing)
L263: cross, not across
L317: species, not species (species is both the singular and plural form)
L441: insect pest outbreaks (pests does not need to be plural because outbreaks is)
Response 2:
All points are changed in the text.

Reviewer 3 Report
This is an interesting review paper, in which the authors report about an impact of climate change on agricultural invasive insect pests in an Old Continent. The topic of the paper is up-to-date and since the paper has all elements of scientific review paper I suggest its publishing; however only after minor revision, since I found some places in the text, which should be improved.
p. 3, line 105: I suggest to mention the following paper: BERGANT et al., 2005. Impact of climate change on developmental dynamics of Thrips tabaci (Thysanoptera: Thripidae) : can it be quantified?. Environmental entomology, 34, 4: 755-766.
p. 6, line 199: the full Latin names of the species should be written according to Fauna Europaea; that means that proper Latin name of tobacco whitefy is "Bemisia tabaci (Gennadius)" and no "Bemisia tabaci Gennadius), proper Latin name of Colorado potato beetle is "Leptinotarsa decemlineata (Say) and no "Leptinotarsa decemlineata Say"..... This should be taken into account in all paper! It is important to mention that when Latin name is written within (plain) brackets - for example (Leptinotarsa decemlineata ...) - then the author(s) of the species should be written within square brackets; i.e. (Leptinotarsa decemlineata [Say]).
p. 10, line 397: replace "polikilotherms" with "poikilotherms"
Author Response
Reviewer #3:
Dear reviewer,
I would like to thank you for your time and effort in reading our article. Also, for your comments and suggestions.
RESPONSE TO REVIEWER’S COMMENTS:
Point 1. p. 3, line 105: I suggest to mention the following paper: BERGANT et al., 2005. Impact of climate change on developmental dynamics of Thrips tabaci (Thysanoptera: Thripidae): can it be quantified?. Environmental entomology, 34, 4: 755-766.
Response 1: Reference added in the text – now L129 – L138
Point 2. p. 6, line 199: the full Latin names of the species should be written according to Fauna Europaea; that means that proper Latin name of tobacco whitefy is "Bemisia tabaci (Gennadius)" and no "Bemisia tabaci Gennadius), proper Latin name of Colorado potato beetle is "Leptinotarsa decemlineata (Say) and no "Leptinotarsa decemlineata Say"..... This should be taken into account in all paper! It is important to mention that when Latin name is written within (plain) brackets - for example (Leptinotarsa decemlineata ...) - then the author(s) of the species should be written within square brackets; i.e. (Leptinotarsa decemlineata [Say]).
Response 2: Points changed in all paper.
Point 3. p. 10, line 397: replace "polikilotherms" with "poikilotherms"
Response 3: Point changed in the text. Now L451

Reviewer 4 Report
Please find the attached manuscript of "The impact of climate change on agricultural invasive insect pests in Europe" I have used track changes for the corrections.

Author Response
Reviewer #4:
Dear reviewer,
I would like to thank you for your time and effort in reading our article. Also, for your comments and suggestions.
RESPONSE TO REVIEWER’S COMMENTS:
Point 1. Please find the attached manuscript of "The impact of climate change on agricultural invasive insect pests in Europe" I have used track changes for the corrections.
Response 1:
The title is changed.
Word ˝major˝ added in Abstract.
L31: Reference added in text. Now L35.
L32: Additional explanation added. Now L36 - L41
L45: Additional explanation added. Now L53 – L58
L52: Point changed in text. Now L67 – L71.
L69: Point changed in text. Now L85
L91: Point changed in text. Now L115
L583: The term is replaced in the text. Now: L640 – L642
L586: Additional explanation on the term ˝re-invasion˝ has been inserted. Now: L645 – L651
L985 – Reference removed
L997 – Reference removed
